# Late Menarche, Not Reproductive Period, Is Associated with Poor Cognitive Function in Postmenopausal Women in Taiwan

**DOI:** 10.3390/ijerph18052345

**Published:** 2021-02-27

**Authors:** Hung-Tse Chou, Pei-Yu Wu, Jiun-Chi Huang, Szu-Chia Chen, Wan-Yi Ho

**Affiliations:** 1Department of General Medicine, Kaohsiung Medical University Hospital, Kaohsiung 807, Taiwan; zhouh1140@gmail.com; 2Department of Internal Medicine, Kaohsiung Municipal Siaogang Hospital, Kaohsiung Medical University, Kaohsiung 812, Taiwan; wpuw17@gmail.com (P.-Y.W.); karajan77@gmail.com (J.-C.H.); scarchenone@yahoo.com.tw (S.-C.C.); 3Division of Nephrology, Department of Internal Medicine, Kaohsiung Medical University Hospital, Kaohsiung Medical University, Kaohsiung 807, Taiwan; 4Faculty of Medicine, College of Medicine, Kaohsiung Medical University, Kaohsiung 807, Taiwan; 5Research Center for Environmental Medicine, Kaohsiung Medical University, Kaohsiung 807, Taiwan; 6Department of Anatomy, School of Medicine, College of Medicine, Kaohsiung Medical University, Kaohsiung 807, Taiwan

**Keywords:** menarche age, reproductive period, cognitive decline, mini mental state exam and subdomains

## Abstract

Female sex hormones such as estrogen and progesterone play an important role in the regulation of a woman’s body, including cognition and neurogenesis. However, the effects of age at menarche and reproductive period on cognitive function are still controversial. The aim of this study was to investigate the relationships between age at menarche and reproductive period with cognitive impairment. Data were obtained from the Taiwan Biobank. Cognitive function was assessed using the Mini Mental State Examination (MMSE) and its five subdomains. Multivariable linear regression analysis revealed that an old age at menarche (per one year; coefficient β, −0.189; *p* = 0.020) was significantly associated with a low total MMSE score, whereas reproductive period (*p* = 0.733) was not significantly associated with total MMSE score. Furthermore, an old age at menarche was significantly associated with low MMSE G2 (registration) (per one year; coefficient β, −0.022; *p* = 0.035) and G5 (language, construction and obey) scores (per one year; coefficient β, −0.054; *p* = 0.047). However, age at menarche was not significantly associated with MMSE G1 (orientation), G3 (attention and calculation) and G4 (recall) scores. In addition, reproductive period was not significantly associated with any MMSE subscores. Late menarche was associated with poor cognitive function, including low total MMSE score and low MMSE G2 and G5 scores. However, reproductive period was not associated with cognitive function in postmenopausal women.

## 1. Introduction

In women, menopause is a common cause of many symptoms and diseases, including hot flushes, mood swings, depression, insomnia, dry vagina, mental confusion, incontinence, osteoporotic symptoms, and vasomotor symptoms [1]. When it comes to treatment of menopausal symptoms, hormone replacement therapy (HRT) is considered the first option to achieve therapeutic relief [2,3]. However, nonhormonal therapy, such as nutraceuticals, can still be useful, despite limited contexts [4]. In a recent prospective observational study, herbal remedy from pollen extracts is superior to soy isoflavones in relieving hot flushes, sleep disturbances and menopause-related symptoms [5].

Still, female sex hormones such as estrogen and progesterone play an important role in the regulation of a woman’s body, and several studies have shown the benefits of female sex hormones including both endogenous and exogenous effects on the central nervous system [6,7,8,9,10,11,12,13,14]. In addition, several studies have demonstrated the significant role of sex steroid hormones, and in particular the effect of estrogen on cognition and progesterone on neurogenesis [15,16,17,18]. Consequently, these findings suggest that conditions such as early menarche, late menopause and longer reproductive period may be associated with neural function [15].

In a literature review, one study showed that late menopause and nulliparity were associated with less cognitive decline [19]. However, in other cohort studies, age at menopause was unrelated to the risk of Alzheimer’s disease [20,21,22,23,24]. The relationship between a longer reproductive period and less cognitive impairment is still controversial [25,26,27,28,29,30]. During reproductive period, estradiol (E2), which is produced during menstrual cycle and before pregnancy and menopause, is the best estrogen to provide neuroprotective effects via both its genomic (receptor-dependent) and nongenomic (receptor-independent) mechanisms [31]. A profound study also found that longer reproductive years was associated with greater telomere length and lower telomerase activity in peripheral blood mononuclear cells [32], noted that short telomeres in leukocytes have been associated in many studies with age-related diseases including cardiovascular disease, Alzheimer’s disease and some cancers [33]. As a result, this suggested that the longer endogenous estrogens exposure may be linked to deceleration of cellular aging. In light of menarche, the relationship between age at menarche and cognitive function is also disputed. One previous study revealed no significant impact on cognitive function in adult women [34], whereas a recent cohort study indicates that later age at menarche was associated with elevated risk of dementia [35]. Other studies have reported that earlier menarche has a protective effect against psychiatric diseases such as depression and schizophrenia [36,37,38].

The aim of this study was therefore to investigate the relationships between age at menarche and reproductive period with cognitive function. We used data from the Taiwan Biobank (TWB) and assessed cognitive function using Mini Mental State Examination (MMSE) total and subdomain scores [39].

## 2. Materials and Methods

### 2.1. Ethics Statement

This study was approved by the Institutional Review Board (IRB) of Kaohsiung Medical University Hospital (KMUHIRB-E(I)-20180242) and conducted according to the principles of the Declaration of Helsinki. The TWB was granted ethical approval by the IRB on Biomedical Science Research/IRB-BM, Academia Sinica, Taiwan, and the Ethics and Governance Council of the TWB, Taiwan. All of the participants provided written informed consent to participate in this study in accordance with institutional requirements.

### 2.2. TWB

The government-supported TWB was created to document lifestyle and genomic data of the residents of Taiwan, and it includes data of volunteers from the community aged 30 to 70 years with no history of cancer [40,41]. All of the volunteers provided blood samples and received in-person interviews where they completed questionnaires and underwent physical examinations. The study is a retrospective study. All of the participants signed informed consent forms. In this study, we included 5000 individuals registered in the TWB up to April 2014.

Data on body height and weight were obtained from the TWB, and body mass index (BMI) was calculated as weight (kg)/height (m)^2^. Data on personal and lifestyle factors including exercise were obtained from the questionnaires. In this study, we defined “exercise” as a leisure activity such as yoga, running, swimming, playing a sport, hiking, cycling, and exercise-based computer games, but occupational activities were not included. Regular exercise was defined as engaging in one of these physical activities over 30 min at least three times per week [42].

### 2.3. Collection of Demographic, Medical and Laboratory Data

The following baseline variables were recorded: demographic features (age and sex), history of tobacco smoking and alcohol consumption, medical history (hypertension, coronary artery disease, diabetes mellitus [DM] and cerebrovascular disease), systolic blood pressure [SBP] and diastolic blood pressure [DBP], and laboratory data (total cholesterol, high-density lipoprotein [HDL] cholesterol, low-density lipoprotein [LDL] cholesterol, fasting glucose, triglycerides, hemoglobin, estimated glomerular filtration rate [eGFR] and uric acid). eGFR was calculated according to the Modification of Diet in Renal Disease four-variable equation [43].

### 2.4. Assessment of Age at Menarche and Menopause

The participants were asked the following questions: ‘How old were you at your first period or menstrual cycle?’, ‘Have your periods stopped completely?’ (‘Yes’ or ‘No’), and ‘How old were you when your periods ceased?’ Reproductive period was calculated as the difference between the age at menopause and the age at menarche. The participants were also asked about whether they used oral contraceptives or HRT for more than six months. Information on birth history, birth times, breastfeeding history, breastfeeding period, contraceptive use history, contraceptive use period and irregular menstrual cycle were also recorded.

### 2.5. Evaluation of Cognitive Function

We assessed the cognitive function of the participants using the MMSE [39]. The MMSE is used as a screening tool for cognitive impairment, and a low score indicates that further evaluations are needed. The MMSE contains five subscales: G1, orientation (score 0–10) (orientation to time, and orientation to place), G2, registration (score 0–3), G3, attention and calculation (score 0–3), G4, recall (score 0–3), and G5, language, construction and obedience (score 0–11) (including reading, repetition, naming, sentence, construction and obedience). The total MMSE score was calculated as the sum of all subscores, with a maximum score of 30. Five hundred and twenty postmenopausal women with complete MMSE measurements during the enrollment period were included in this study (Figure 1).

### 2.6. Statistical Analysis

Statistical analysis was performed using SPSS version 19.0 for Windows (SPSS Inc. Chicago, IL, USA). Data are presented as number (percentage), mean (standard deviation), or median (25th–75th percentile) for triglycerides. A MMSE cut-off score of 24 was used to classify the severity of cognitive impairment. The chi-square test was used to analyze between-groups differences in categorical variables, and the independent t test was used for continuous variables. Multiple comparisons among the study participants according to the age at menarche were performed using one-way analysis of variance. Linear regression analysis was used to identify associations between age at menarche and reproductive period and the MMSE and its subscales. Significant variables in the univariable analysis, age at menarche and reproductive period were selected into the multivariable analysis. A *p* value of less than 0.05 was considered to indicate a statistically significant difference.

## 3. Results

The mean age of the 520 female participants was 63.7 ± 2.9 years. The participants were stratified into two groups according to MMSE ≥ 24 (n = 445, 85.6%) or < 24 (n = 75, 14.4%). A comparison of the clinical characteristics between these two groups is shown in Table 1. Compared to the participants with MMSE ≥ 24, those with MMSE < 24 were older, more had an education level of higher than senior high school, higher rate of employment and higher BMI. A comparison of MMSE and MMSE subscores and female menstruation related conditions between these two groups is shown in Table 2. Compared to the participants with MMSE ≥ 24, those with MMSE < 24 had lower scores of each MMSE subdomain as well as total MMSE score, higher menarche age, higher birth times and higher breastfeeding period.

### 3.1. Association between MMSE Total Score and SubScores According to Age at Menarche

Table 3 shows the MMSE total and subdomain scores according to age at menarche. There were significant trends of stepwise decreases in MMSE G1 (*p* = 0.011), G3 (*p* = 0.030), G5 (*p* = 0.010) and total MMSE (*p* = 0.001) scores.

### 3.2. Determinants of Total MMSE Score

Table 4 shows the determinants of total MMSE score in the study participants using linear regression analysis. In the univariable analysis, older age (per one year; coefficient β, −0.151; *p* = 0.001), cerebrovascular disease (coefficient β, −3.306; *p* = 0.025), education level lower than senior high school (coefficient β, 2.252; *p* < 0.001), not living alone (coefficient β, 0.750; *p* = 0.040), high BMI (per 1 kg/m^2^; coefficient β, −0.166; *p* < 0.001), low total cholesterol (per 1 mg/dL; coefficient β, 0.007; *p* = 0.037), low LDL-cholesterol (per 1 mg/dL; coefficient β, 0.008; *p* = 0.030), old age at menarche (per one year; coefficient β, −0.347; *p* < 0.001), high birth times (per 1 time; coefficient β, −0.667; *p* < 0.001), and high breastfeeding period (per one month; coefficient β, −0.035; *p* < 0.001) were associated with low total MMSE score. After adjusting for age, cerebrovascular disease, education level, living alone, BMI, total cholesterol, LDL-cholesterol, age at menarche, reproductive period, breath times and breastfeeding period, education level lower than senior high school (coefficient β, 1.591; *p* < 0.001), high BMI (per 1 kg/m^2^; coefficient β, −0.103; *p* = 0.011), and old age at menarche (per one year; coefficient β, −0.189; *p* = 0.020) were significantly associated with low total MMSE score, whereas reproductive period was not significantly associated with total MMSE score (*p* = 0.733).

### 3.3. Correlations between Age at Menarche and Reproductive Period and Each MMSE Subdomain

Table 5 shows the associations between age at menarche and reproductive period with MMSE subscores in the study participants using multivariable linear regression analysis. The results showed that older age at menarche was significantly associated with low MMSE G2 (per one year; coefficient β, −0.022; *p* = 0.035) and G5 (per one year; coefficient β, −0.054; *p* = 0.047) scores. However, age at menarche was not significantly associated with MMSE G1, G3 and G4 scores. In addition, reproductive period was not significantly associated with any of the MMSE subdomains.

## 4. Discussion

The results of this study indicate that in postmenopausal women, older age at menarche was associated with a low total MMSE score, low G2 score (registration), and low G5 score (language, construction and obedience). However, no statistical significance was found in the relationship between reproductive period and MMSE scores.

The first important finding of this study is that late menarche was associated with poor cognitive function as assessed using the MMSE, including low total MMSE score and low MMSE G2 and G5 scores, not shorter reproductive period. An experiment in female mice suggests that pubertal hormones are critical for the maturation of the frontal cortex [44]. The researchers emphasize that prepubertal, but not postpubertal, gonadectomy blocks the increase in inhibitory neurotransmission. Actually, estrogen is served as a multipurpose brain messenger with direct and indirect effects on estrogen receptors concentrated in several areas of the brain [14]. In studies, membrane-associated estrogen receptors are observed in the prefrontal cortex, dorsal striatum, nucleus accumbens, and hippocampus, all of which are involved in higher brain functions, such as motor planning, decision-making, learning, and memory [45,46,47]. As we understood, prefrontal cortex being part of the frontal cortex is in charge of many cerebral functions, including language, executive functions, emotional behavior, temporal integration, working memory, etc. [48]. All cognitive functions in the prefrontal cortex seem to reach a relative plateau of maturity at about the age of 12 years. However, higher cognitive functions such as language and intelligence continue to develop into the third decade of life [48]. As a result, we considered earlier age at menarche as promoting the maturation of certain cognitive functions in this critical time period, and this has nothing to do with longer reproductive years. Although the results of the two factors which affect more the cognitive function remain inconsistent among studies, it is possible that an earlier age at menarche and exposure to gonadal hormones may subsequently affect cognitive function in later life independently of its effect on extending the reproductive period.

For MMSE subdomains, G2 and G5 involving abilities consisting of registration, language (containing reading, repetition, naming, sentence, and obedience) and visual construction present decline in group with later menarche [49,50]. Based on current evidences in the previous paragraph, we suggest that the maturation process of the prefrontal area could partially explain this phenomenon, especially better language function carried out by the prefrontal cortex in the group with earlier menarche. In a study of 10 to 15 year-olds, after controlling for age, estradiol levels were related to higher grey matter density in the middle frontal, inferior temporal, and middle occipital gyrus [51]. Focusing on the inferior temporal gyrus, this region where visual-object processing finally culminates is associated with the representation of objects, places, faces, and colors [52]. Thus, higher scores in visual construction part could contribute to better visual ability from the inferior temporal gyrus and superior motor capability from the prefrontal cortex. However, to date, possible reasons behind registration in the G2 subdomain remain unknown due to scarce data. Further research is expected to figure out the question.

In addition, several studies have observed related outcomes, through exploration of exogenous estrogen and HRT effects on cognition. A double-blind clinical trial reported that a transient increase in the concentration of plasma estrogen in postmenopausal women significantly improved cognitive function related to the prefrontal cortex as assessed using a digit ordering task which required the short-term memory, whereas memory associated with the hippocampus was less affected [12]. Another study compared the findings from behavioral and neuroimaging studies on nonhuman primates and humans, and found that estrogen may influence working memory tasks mediated by both the prefrontal cortex and hippocampus [10]. In a double-blind trial, compared to treatment with a placebo, estrogen treatment reduced perseveration errors during verbal recall, a process mediated by the frontal system, but did not affect other cognitive processes [11]. In addition, a randomized control trial reported that estrogen replacement therapy enhanced verbal memory (the ability to recognize or remember a verbal stimulus or association over a certain period) and learning in postmenopausal women, but that it had no effect on spatial ability or visual memory [8]. Another short-term clinical trial suggested that hormone therapy may enhance verbal memory after surgical menopause. However, observational studies have reported no substantial effect of natural menopause or estrogen-containing hormone therapy on episodic memory [9]. These findings provide more clues in explaining the association between age at menarche and cognitive function, and also indicate that women with early menarche may benefit from the neuroprotective effects of estrogen.

In the present study, we found that high BMI is associated with low MMSE scores. Elderly women with higher BMI have higher plasma concentrations of estrogens [53] producing by aromatase from adipose tissue [54], so we may postulate that they could take greater advantage of female sex hormones. However, according to a systemic literature review, their findings provide evidence that midlife obese individuals exhibit cognitive problems in the following domains: intellectual functioning, psychomotor performance and speed, visual construction, concept formation and set-shifting, and decision-making [55]. This review is consistent with our findings. The estrogenic benefit of high BMI is balanced by the fact that higher adiposity is associated with decreased cardiovascular health and decreased insulin sensitivity, both of which increase AD risk [56,57].

There are some limitations to this study. First, we could not examine the exact causes of a decline in MMSE scores as this was a cross-sectional study. Follow-up studies are needed to confirm our results. Second, the participants may have been subject to recall bias, especially those with poor cognitive function. Third, cognitive performance was assessed using only the MMSE and its subscales, which may have affected the results. Although the MMSE has some limitations, its use as a brief screening test to quantitatively evaluate the severity of cognitive impairment and document changes over time has been validated [58].

In conclusion, late menarche was associated with worse cognitive function, including low total MMSE score and low MMSE G2 and G5 scores. However, reproductive period was not associated with cognitive function in postmenopausal women. We hypothesize that women with early menarche benefit from the neuroprotective effects of estrogen with an earlier exposure to gonadal hormones.

## Figures and Tables

**Figure 1 ijerph-18-02345-f001:**
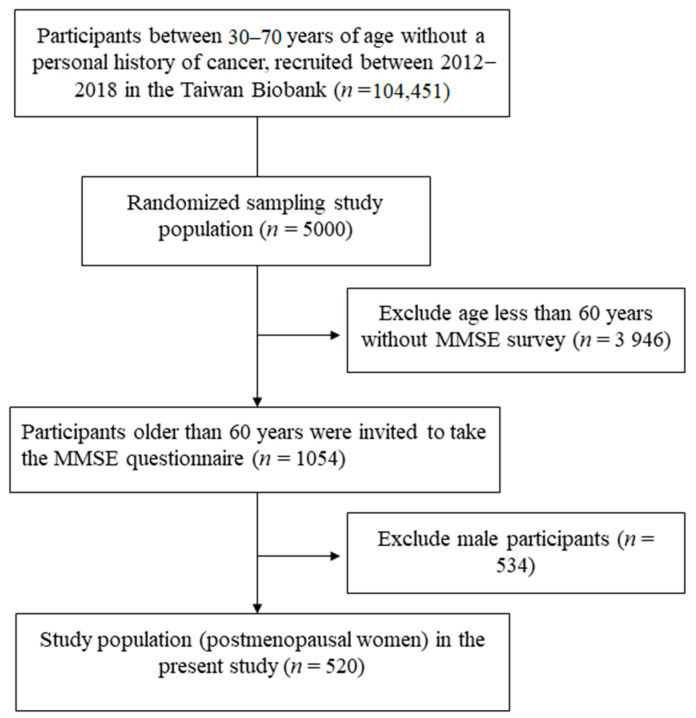
Flowchart of study population.

**Table 1 ijerph-18-02345-t001:** Comparison of clinical characteristics among participants according to total Mini Mental State Examination (MMSE) scores ≥ 24 or < 24.

Characteristics	All (*n* = 520)	MMSE ≥ 24 (*n* = 445)	MMSE < 24 (*n* = 75)	*p*
Age (year)	63.7 ± 2.9	63.6 ± 2.9	64.3 ± 3.0	0.046
Smoking history (%)	3.7	3.8	2.7	1.000
Alcohol history (%)	0.4	0.4	0	1.000
DM (%)	16.3	16.4	16.0	0.930
Hypertension (%)	20.8	20.2	24.0	0.456
Coronary artery disease (%)	2.5	2.5	2.7	1.000
Cerebrovascular disease (%)	0.8	0.7	1.3	0.465
Education higher than senior high schools (%)	56.0	62.7	16.0	<0.001
Living alone (%)	14.6	15.5	9.3	0.162
Having job (%)	18.8	17.4	27.5	0.046
Regular exercise habits (%)	68.1	67.4	72.0	0.431
Midnight snack habits (%)	16.7	16.9	16.0	0.855
BMI (kg/m^2^)	24.0 ± 3.3	23.9 ± 3.2	24.8 ± 3.6	0.016
SBP (mmHg)	124.2 ± 17.7	123.8 ± 17.6	126.7 ± 18.3	0.190
DBP (mmHg)	70.0 ± 10.3	69.9 ± 10.5	70.2 ± 9.2	0.846
Laboratory parameters				
Fasting glucose (mg/dL)	99.7 ± 22.0	99.3 ± 20.8	102.0 ± 27.8	0.327
Triglyceride (mg/dL)	100 (76–133)	99 (75–133)	102 (79–134)	0.967
Total cholesterol (mg/dL)	210.0 ± 36.4	211.0 ± 63.7	204.2 ± 34.5	0.133
HDL-cholesterol (mg/dL)	58.3 ± 13.4	58.2 ± 13.4	58.8 ± 13.4	0.743
LDL-cholesterol (mg/dL)	130.1 ± 33.4	130.9 ± 33.8	124.9 ± 30.6	0.148
Hemoglobin (g/dL)	13.3 ± 1.0	13.3 ± 1.0	13.4 ± 1.1	0.365
eGFR (mL/min/1.73 m^2^)	107.6 ± 24.9	107.8 ± 24.9	106.1 ± 25.1	0.589
Uric acid (mg/dL)	5.2 ± 1.2	5.1 ± 1.2	5.2 ± 1.2	0.523

Abbreviations. MMSE, Mini Mental State Examination; DM, diabetes mellitus; BMI, body mass index; SBP, systolic blood pressure; DBP, diastolic blood pressure; HDL, high-density lipoprotein; LDL, low-density lipoprotein; eGFR, estimated glomerular filtration rate.

**Table 2 ijerph-18-02345-t002:** Comparison of MMSE and MMSE subscores and female menstruation related conditions among participants according to total MMSE scores ≥ 24 or < 24.

Characteristics	All (*n* = 520)	MMSE ≥ 24 (*n* = 445)	MMSE < 24 (*n* = 75)	*p*
MMSE				
G1 (Orientation)	9.4 ± 0.9	9.6 ± 0.7	8.5 ± 1.3	<0.001
G2 (Registration)	2.9 ± 0.3	3.0 ± 0.3	2.8 ± 0.5	0.076
G3 (Attention & Calculation)	3.6 ± 1.8	3.9 ± 1.6	1.3 ± 1.1	<0.001
G4 (Recall)	2.2 ± 0.9	2.4 ± 0.8	1.3 ± 1.1	<0.001
G5 (Language, construction & obey)	8.4 ± 0.9	8.6 ± 0.7	7.2 ± 1.3	<0.001
MMSE total	26.5 ± 2.9	27.4 ± 2.0	21.2 ± 1.9	<0.001
Menstruation related conditions				
Age of menarche (year)	14.3 ± 1.6	14.2 ± 1.5	15.1 ± 1.9	<0.001
Reproductive period (years)	36.0 ± 5.2	36.2 ± 5.2	35.0 ± 5.4	0.070
Irregular menstrual cycle (%)	14.3	14.8	10.8	0.360
Birth history (%)	99.4	99.5	98.6	0.373
Birth times	2.7 ± 1.0	2.7 ± 1.0	3.1 ± 1.1	0.005
Breast feeding history (%)	75.3	83.1	74.0	0.100
Breastfeeding period (months)	16.2 ± 19.6	15.0 ± 19.5	23.5 ± 18.5	0.001
Hormone therapy history (%)	34.9	35.4	32.0	0.563
Contraceptive use history (%)	3.1	3.4	1.3	0.489
Contraceptive use period (years)	4.5 ± 1.2	4.3 ± 6.5	6.0 ± 0.0	0.809

Abbreviations. MMSE, mini-mental state examination.

**Table 3 ijerph-18-02345-t003:** Comparison of MMSE total and subscores according to age of menarche.

	Menarche Age (Year)	≦12 (*n* = 54)	13 (*n* = 104)	14 (*n* = 162)	15 (*n* = 82)	16 (*n* = 68)	≧17 (*n* = 50)	*p*
MMSE								
G1 (Orientation)		9.5 ± 0.8	9.4 ± 0.8	9.6 ± 0.7	9.5 ± 0.9	9.3 ± 1.1	9.1 ± 1.1	0.011
G2 (Registration)		3.0 ± 0.0	3.0 ± 0.3	2.9 ± 0.3	2.9 ± 0.4	2.9 ± 0.4	2.8 ± 0.5	0.110
G3 (Attention &Calculation)		3.6 ± 1.7	3.7 ± 1.8	3.8 ± 1.7	3.5 ± 1.7	2.9 ± 1.9	3.4 ± 2.0	0.030
G4 (Recall)		2.3 ± 0.7	2.3 ± 0.9	2.3 ± 0.9	2.2 ± 0.9	2.3 ± 1.0	1.9 ± 1.0	0.190
G5 (Language, construction & obey)		8.5 ± 0.8	8.6 ± 0.7	8.3 ± 1.0	8.4 ± 1.0	8.2 ± 1.1	8.1 ± 1.0	0.010
MMSE total		26.9 ± 2.5	26.9 ± 2.7	26.9 ± 2.6	26.6 ± 3.0	25.6 ± 3.2	25.3 ± 3.8	0.001

Values are expressed as mean ± standard deviation.

**Table 4 ijerph-18-02345-t004:** Determinants total MMSE scores using linear regression analysis.

Characteristics	Univariable		Multivariable	
	Coefficient β (95% CI)	*p*	Coefficient β (95% CI)	*p*
Age (per one year)	−0.151 (−0.238, −0.064)	0.001	−0.040 (−0.127, 0.047)	0.371
Smoking history (ever vs. never)	0.105 (−1.246, 1.455)	0.879	-	-
Alcohol history (ever vs. never)	−0.533 (−4.626, 3.561)	0.798	-	-
DM	−0.255 (−0.940, 0.430)	0.465	-	-
Hypertension	−0.483 (−1.106, 0.140)	0.129	-	-
Coronary artery disease	−0.229 (−1.852, 1.394)	0.782	-	-
Cerebrovascular disease	−3.306 (−6.193, −0.420)	0.025	−2.013 (−4.617, 0.591)	0.129
Education higher than senior high schools	2.252 (1.780, 2.724)	<0.001	1.591 (1.052, 2.130)	<0.001
Living alone	0.750 (0.035, 1.464)	0.040	0.585 (−0.122, 1.292)	0.105
Having job	−0.335 (−0.983, 0.314)	0.311	-	-
Regular exercise habits	−0.326 (−0.869, 0.216)	0.238	-	-
Midnight snack habits	0.053 (−0.626, 0.732)	0.879	-	-
BMI (per 1 kg/m^2^)	−0.166 (−0.242, −0.089)	<0.001	−0.103 (−0.183, −0.023)	0.011
SBP (per 1 mmHg)	−0.007 (−0.021, 0.008)	0.350	-	-
DBP (per 1 mmHg)	0.002 (−0.023, 0.026)	0.889	-	-
Laboratory parameters				
Fasting glucose (per 1 mg/dL)	−0.011 (−0.022, 0.001)	0.066	-	-
Triglyceride (log per 1mg/dL)	−0.718 (−2.025, 0.589)	0.281	-	-
Total cholesterol (per 1 mg/dL)	0.007 (0, 0.014)	0.037	−0.005 (−0.020, 0.009)	0.448
HDL-cholesterol (per 1 mg/dL)	0.003 (−0.016, 0.022)	0.759	-	-
LDL-cholesterol (per 1 mg/dL)	0.008 (0.001, 0.016)	0.030	0.013 (−0.002, 0.028)	0.094
Hemoglobin (per 1 g/dL)	−0.085 (−0.328, 0.159)	0.496	-	-
eGFR (per 1 mL/min/1.73 m^2^)	0.002 (−0.008, 0.012)	0.741	-	-
Uric acid (per 1 mg/dL)	−0.159 (−0.366, 0.048)	0.131	-	-
Menstruation related conditions				
Age of menarche (per one year)	−0.347 (−0.501, −0.192)	<0.001	−0.189 (−0.348, −0.030)	0.020
Reproductive period (per one year)	0.045 (−0.004, 0.094)	0.069	0.008 (−0.040, 0.056)	0.733
Menstrual cycle (irregular vs. regular)	0.193 (−0.524, 0.910)	0.598	-	-
Birth history	0.170 (−3.161, 3.500)	0.920	-	-
Birth times (per one time)	−0.667 (−0.922, −0.412)	<0.001	−0.121 (−0.404, 0.162)	0.402
Breastfeeding period (per one month)	−0.035 (−0.048, −0.023)	<0.001	−0.007 (−0.022, 0.099)	0.398
Hormone therapy history	0.037 (−0.496, 0.570)	0.893	-	-
Contraceptive use history	0.041 (−1.428, 1.510)	0.957	-	-
Contraceptive use period (per one year)	−0.131 (−0.377, 0.115)	0.269	-	-

Values expressed as unstandardized coefficient β and 95% confidence interval (CI). Abbreviations are same as Table 1. Adjusted for age, cerebrovascular disease, education level, living alone, BMI, total cholesterol, LDL-cholesterol, menarche age, reproductive period, breath times and breastfeeding period.

**Table 5 ijerph-18-02345-t005:** Association of menarche age and reproductive period with MMSE subscores using multivariable linear regression analysis.

Characteristics	Multivariable	
	Coefficient β (95% CI)	*p*
G1 (Orientation)		
Age of menarche (per one year)	−0.028 (−0.078, 0.023)	0.281
Reproductive period (per one year)	0.006 (−0.009, 0.022)	0.403
G2 (Registration)		
Age of menarche (per one year)	−0.022 (−0.042, −0.002)	0.035
Reproductive period (per one year)	0 (−0.006, 0.007)	0.904
G3 (Attention &Calculation)		
Age of menarche (per one year)	−0.073 (−0.178, 0.032)	0.174
Reproductive period (per one year)	0.001 (−0.030, 0.033)	0.937
G4 (Recall)		
Age of menarche (per one year)	−0.013 (−0.066, 0.040)	0.637
Reproductive period (per one year)	−0.003 (−0.019, 0.013)	0.693
G5 (Language, construction and obey)		
Age of menarche (per one year)	−0.054 (−0.107, 0)	0.047
Reproductive period (per one year)	0.003 (−0.013, 0.020)	0.674

Values expressed as unstandardized coefficient β and 95% confidence interval (CI). Adjusted for age, cerebrovascular disease, education level, living alone, BMI, total cholesterol, LDL-cholesterol, menarche age, reproductive period, breath times and breastfeeding period.

## Data Availability

The data underlying this study is from the Taiwan Biobank. Due to restrictions placed on the data by the Personal Information Protection Act of Taiwan, the minimal data set cannot be made publicly available. Data may be available upon request to interested researchers. Please send data requests to: Szu-Chia Chen, PhD, MD. Division of Nephrology, Department of Internal Medicine, Kaohsiung Medical University Hospital, Kaohsiung Medical University.

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
