# Peer review of "Late Menarche, Not Reproductive Period, Is Associated with Poor Cognitive Function in Postmenopausal Women in Taiwan"

_ijerph, 2021, doi:10.3390/ijerph18052345_

Round 1
Reviewer 1 Report
I consider this work to be very original and interesting. It is well written and well understood. The analyzes carried out seem very relevant to me
I make the following comments:
1. In the introduction I might have liked a more in-depth look at estrogen exposure during reproductive age. Have women who have taken contraceptives been taken into account and how long they have taken it? On the other hand, is there a difference between exposure to natural or artificial estrogens (for example, that provided by contraceptive pills)?
2. In material and methods: The number of pregnancies and if there was breastfeeding and for how long have not been taken into account either. I believe that all these data can be useful in estimating estrogen exposure during the fertile life of the patient and should have been taken into account.
3. Regarding the discussion, the authors will be able to argue why they have only found differences in 2 sub-scales of the test.
4. For the bibliography in general, it is quite old. There is only one reference from 2019. Perhaps the authors could update it.
5. for the tables, the letter of the flow chart is somewhat small.
Author Response
Reviewer 1
I consider this work to be very original and interesting. It is well written and well understood. The analyzes carried out seem very relevant to me. I make the following comments:
- In the introduction I might have liked a more in-depth look at estrogen exposure during reproductive age. Have women who have taken contraceptives been taken into account and how long they have taken it? On the other hand, is there a difference between exposure to natural or artificial estrogens (for example, that provided by contraceptive pills)?
- Ans: Thank you for your comments. First, we have added more about in-depth look at estrogen exposure during reproductive age in Introduction. Besides, we have added the data of contraceptive use history and contraceptive use period in Table 2, and further analyzed in Table 4. Contraceptive use history or period were noted associated with MMSE.
- During reproductive period, estradiol (E2), which is produced during menstrual cycle and before pregnancy and menopause, is the best estrogen to provide neuroprotective effects via both its genomic (receptor-dependent) and nongenomic (receptor-independent) mechanisms [31]. A profound study also found that longer reproductive years was associated with greater telomere length and lower telomerase activity in peripheral blood mononuclear cells [32], noted that short telomeres in leukocytes have been associated in many studies with age-related diseases including cardiovascular disease, Alzheimer’s disease and some cancers [33]. As a result, this suggested that the longer endogenous estrogens exposure may be linked to deceleration of cellular aging. In light of menarche, the relationship between age at menarche and cognitive function is also disputed. One previous study revealed no significant impact on cognitive function in adult women [34], whereas a recent cohort study indicates that later age at menarche was associated with elevated risk of dementia [35]. (Page 3, Line 23 to Page 4, Line 10)
- In material and methods: The number of pregnancies and if there was breastfeeding and for how long have not been taken into account either. I believe that all these data can be useful in estimating estrogen exposure during the fertile life of the patient and should have been taken into account.
- Ans: Thank you for your comments. We totally agreed that pregnancy numbers and breastfeeding history and period are important in estimating estrogen exposure. We have added the data of birth times, breastfeeding history and period in Table 2, and further analyzed in Table 4.
- Regarding the discussion, the authors will be able to argue why they have only found differences in 2 sub-scales of the test.
- Ans: Thank you for your comments. We have added the issue in the Discussion.
- The first important finding of this study is that late menarche was associated with poor cognitive function as assessed using the MMSE, including low total MMSE score and low MMSE G2 and G5 scores, not shorter reproductive period. An experiment in female mice suggests that pubertal hormones are critical for the maturation of the frontal cortex [44]. The researchers emphasize that pre-pubertal, but not post-pubertal, gonadectomy blocks the increase in inhibitory neurotransmission. Actually, estrogen is served as a multipurpose brain messenger with direct and indirect effects on estrogen receptors concentrated in several areas of the brain [14]. In studies, membrane-associated estrogen receptors are observed in the prefrontal cortex, dorsal striatum, nucleus accumbens, and hippocampus, all of which are involved in higher brain functions, such as motor planning, decision-making, learning, and memory [45-47]. As we understood, pre-frontal cortex being part of the frontal cortex is in charge of many cerebral functions, including language, executive functions, emotional behavior, temporal integration, working memory, etc [48]. All of cognitive functions in prefrontal cortex seem to reach a relative plateau of maturity at about the age of 12 years. However, higher cognitive functions such as language and intelligence continue to develop into the third decade of life [48]. As a result, we considered earlier age at menarche promoting the maturation of certain cognitive functions in this critical time period, and this has nothing to do with longer reproductive years. Although the results of the two factors which affect more on cognitive function remain inconsistent among studies, it is possible that an earlier age at menarche and exposure to gonadal hormones may subsequently affect cognitive function in later life independently of its effect on extending the reproductive period.
For MMSE subdomains, G2 and G5 involving abilities consisting of registration, language (containing reading, repetition, naming, sentence, and obedience) and visual construction present decline in group with later menarche [49, 50]. Based on current evidences in previous paragraph, we suggest that the maturation process of prefrontal area could partially explain this phenomenon, especially better language function carrying out by prefrontal cortex in group with earlier menarche. In a study of 10 to 15 year-olds, after controlling for age, estradiol levels were related to higher grey matter density in the middle frontal, inferior temporal, and middle occipital gyrus [51]. Focus on inferior temporal gyrus, this region where visual-object processing finally culminates is associated with the representation of objects, places, faces, and colors [52]. Thus, higher scores in visual construction part could contribute to better visual ability from inferior temporal gyrus and superior motor capability from prefrontal cortex. However, to date, possible reasons behind registration in G2 subdomain remain unknown due to scarce data. Further researches are expected to figure out the question. (Page 14, Line 7 to Page 15, Line 18)
- For the bibliography in general, it is quite old. There is only one reference from 2019. Perhaps the authors could update it.
- Ans: Thank you for your comments. We have added some new references.
- for the tables, the letter of the flow chart is somewhat small.
- Ans: Thank you for your suggestion. We have enlarged the letter.
Reviewer 2 Report
the manuscript is interesting, well developed. my suggestions are;
1. put in the journal format
2. what is the bibliographic reference for the following sentence; Regular exercise was defined as engaging in one of these physical
activities for at least 30 minutes a day
3. put the tables where they are cited in the text.
4. Can you expand the discussion? there is a lot of information in the results that is not discussed
5. Table 1, is it too big to divide it into two tables?
Author Response
Reviewer 2
The manuscript is interesting, well developed. my suggestions are;
- put in the journal format.
- Ans: Thank you for your comments. We have put the tables and figure into the manuscript. The journal will help change the format of the article later.
- what is the bibliographic reference for the following sentence; Regular exercise was defined as engaging in one of these physical activities for at least 30 minutes a day
- Ans: Thank you for your comments. We have added the reference.
- Regular exercise was defined as engaging in one of these physical activities over 30 minutes at least three times per week [42]. (Page 5, Line 12-13)
- put the tables where they are cited in the text.
- Ans: Thank you for your comments. We have put the tables where they are cited in the text.
- Can you expand the discussion? there is a lot of information in the results that is not discussed.
- Ans: Thank you for your comments. We have added more content, such as why only differences in 2 sub-scales, BMI and MMSE.
- For MMSE subdomains, G2 and G5 involving abilities consisting of registration, language (containing reading, repetition, naming, sentence, and obedience) and visual construction present decline in group with later menarche [49, 50]. Based on current evidences in previous paragraph, we suggest that the maturation process of prefrontal area could partially explain this phenomenon, especially better language function carrying out by prefrontal cortex in group with earlier menarche. In a study of 10 to 15 year-olds, after controlling for age, estradiol levels were related to higher grey matter density in the middle frontal, inferior temporal, and middle occipital gyrus [51]. Focus on inferior temporal gyrus, this region where visual-object processing finally culminates is associated with the representation of objects, places, faces, and colors [52]. Thus, higher scores in visual construction part could contribute to better visual ability from inferior temporal gyrus and superior motor capability from prefrontal cortex. However, to date, possible reasons behind registration in G2 subdomain remain unknown due to scarce data. Further researches are expected to figure out the question. (Page 15, Line 4-18)
- In the present study, we found that high BMI is associated with low MMSE scores. Elderly women with higher BMI have higher plasma concentrations of estrogens [53] producing by aromatase from adipose tissue [54], so we may postulate that they could take more advantages of female sex hormones. However, according to a systemic literature review, their findings provide evidence that mid-life obese individuals exhibit cognitive problems in the following domains: intellectual functioning, psychomotor performance and speed, visual construction, concept formation and set shifting, and decision making [55]. This review is consistent with our findings. The estrogenic benefit of high BMI is balanced by the fact that higher adiposity is associated with decreased cardiovascular health and decreased insulin sensitivity, both of which increase AD risk [56, 57]. (Page 16, Line 15-25)
- Table 1, is it too big to divide it into two tables?
Ans: Thank you for your suggestion. We have divided Table 1 into two tables.
Reviewer 3 Report
ijerph-1108549
The article deals with the association between cerebral function and late menarche in menopausal women. Authors concluded that late menarche was associated with poor cognitive function.
The topic of this manuscript is interesting and falls within the scope of the journal. I would recommend the following major revisions, to improve the article:
- All the text requires an accurate language revision by an English speaker, in order to correct typos and improve style.
- Introduction: Nutraceutical compounds are widely adopted for their helpful effects on menopausal complaints. Authors should deepen this aspect in the introduction, referring to PMID: 31466381; PMID: 32693763
- Materials and methods: was it a retrospective study? Authors should explain this aspect in the first sentence of the section.
- Materials and methods: was this study designed according to the STROBE Statement (PMID: 18064739). Authors should report this information.
- Results: Please Authors consider to shorten “unstandardized coefficient”
Author Response
Reviewer 3
The article deals with the association between cerebral function and late menarche in menopausal women. Authors concluded that late menarche was associated with poor cognitive function. The topic of this manuscript is interesting and falls within the scope of the journal. I would recommend the following major revisions, to improve the article:
- All the text requires an accurate language revision by an English speaker, in order to correct typos and improve style.
Ans: Thank you for your suggestion. Before submission, we had sent the manuscript to English editing. We have provided English editing certificate as below.
- Introduction: Nutraceutical compounds are widely adopted for their helpful effects on menopausal complaints. Authors should deepen this aspect in the introduction, referring to PMID: 31466381; PMID: 32693763
Ans: Thank you for your comments. We have added the issue in the Introductiion.
- When it comes to treatment of menopausal symptoms, hormones replacement therapy (HRT) is considered the first option to achieve therapeutic relief [2, 3]. However, nonhormonal therapy, such as nutraceuticals, can still be useful, despite limited contexts [4]. In a recent prospective observational study, herbal remedy from pollen extracts is superior to soy isoflavones in relieving hot flushes, sleep disturbances and menopause-related symptoms [5]. (Page 3, Line 4-10)
- Materials and methods: was it a retrospective study? Authors should explain this aspect in the first sentence of the section.
Ans: Yes, the cross-sectional study is a retrospective study. We have added in the Methods.
- The study is a retrospective study. (Page 5, Line 4-5)
- Materials and methods: was this study designed according to the STROBE Statement (PMID: 18064739). Authors should report this information.
Ans: We have added STROBE statement as below and supplement.
STROBE Statement—checklist of items that should be included in reports of observational studies
|
|
Item No |
Recommendation |
Page number |
|
Title and abstract |
1 |
(a) Indicate the study’s design with a commonly used term in the title or the abstract |
1-2 |
|
(b) Provide in the abstract an informative and balanced summary of what was done and what was found |
1-2 |
||
|
Introduction |
|
||
|
Background/rationale |
2 |
Explain the scientific background and rationale for the investigation being reported |
3-4 |
|
Objectives |
3 |
State specific objectives, including any prespecified hypotheses |
3-4 |
|
Methods |
|
||
|
Study design |
4 |
Present key elements of study design early in the paper |
4-7 |
|
Setting |
5 |
Describe the setting, locations, and relevant dates, including periods of recruitment, exposure, follow-up, and data collection |
4-7 |
|
Participants |
6 |
(a) Cohort study—Give the eligibility criteria, and the sources and methods of selection of participants. Describe methods of follow-up Case-control study—Give the eligibility criteria, and the sources and methods of case ascertainment and control selection. Give the rationale for the choice of cases and controls Cross-sectional study—Give the eligibility criteria, and the sources and methods of selection of participants |
4-7 |
|
(b) Cohort study—For matched studies, give matching criteria and number of exposed and unexposed Case-control study—For matched studies, give matching criteria and the number of controls per case |
|
||
|
Variables |
7 |
Clearly define all outcomes, exposures, predictors, potential confounders, and effect modifiers. Give diagnostic criteria, if applicable |
4-7 |
|
Data sources/ measurement |
8* |
For each variable of interest, give sources of data and details of methods of assessment (measurement). Describe comparability of assessment methods if there is more than one group |
4-7 |
|
Bias |
9 |
Describe any efforts to address potential sources of bias |
4-7 |
|
Study size |
10 |
Explain how the study size was arrived at |
4-7 |
|
Quantitative variables |
11 |
Explain how quantitative variables were handled in the analyses. If applicable, describe which groupings were chosen and why |
4-7 |
|
Statistical methods |
12 |
(a) Describe all statistical methods, including those used to control for confounding |
7-8 |
|
(b) Describe any methods used to examine subgroups and interactions |
7-8 |
||
|
(c) Explain how missing data were addressed |
7-8 |
||
|
(d) Cohort study—If applicable, explain how loss to follow-up was addressed Case-control study—If applicable, explain how matching of cases and controls was addressed Cross-sectional study—If applicable, describe analytical methods taking account of sampling strategy |
4-7 |
||
|
(e) Describe any sensitivity analyses |
7-8 |
||
|
Results |
|
||
|
Participants |
13* |
(a) Report numbers of individuals at each stage of study—eg numbers potentially eligible, examined for eligibility, confirmed eligible, included in the study, completing follow-up, and analysed |
8-13 |
|
(b) Give reasons for non-participation at each stage |
8-13 |
||
|
(c) Consider use of a flow diagram |
7 |
||
|
Descriptive data |
14* |
(a) Give characteristics of study participants (eg demographic, clinical, social) and information on exposures and potential confounders |
8-13 |
|
(b) Indicate number of participants with missing data for each variable of interest |
8-13 |
||
|
(c) Cohort study—Summarise follow-up time (eg, average and total amount) |
8-13 |
||
|
Outcome data |
15* |
Cohort study—Report numbers of outcome events or summary measures over time |
8-13 |
|
Case-control study—Report numbers in each exposure category, or summary measures of exposure |
|
||
|
Cross-sectional study—Report numbers of outcome events or summary measures |
|
||
|
Main results |
16 |
(a) Give unadjusted estimates and, if applicable, confounder-adjusted estimates and their precision (eg, 95% confidence interval). Make clear which confounders were adjusted for and why they were included |
8-13 |
|
(b) Report category boundaries when continuous variables were categorized |
8-13 |
||
|
(c) If relevant, consider translating estimates of relative risk into absolute risk for a meaningful time period |
8-13 |
||
|
Other analyses |
17 |
Report other analyses done—eg analyses of subgroups and interactions, and sensitivity analyses |
8-13 |
|
Discussion |
|
||
|
Key results |
18 |
Summarise key results with reference to study objectives |
19 |
|
Limitations |
19 |
Discuss limitations of the study, taking into account sources of potential bias or imprecision. Discuss both direction and magnitude of any potential bias |
14-17 |
|
Interpretation |
20 |
Give a cautious overall interpretation of results considering objectives, limitations, multiplicity of analyses, results from similar studies, and other relevant evidence |
15 |
|
Generalisability |
21 |
Discuss the generalisability (external validity) of the study results |
17 |
|
Other information |
|
||
|
Funding |
22 |
Give the source of funding and the role of the funders for the present study and, if applicable, for the original study on which the present article is based |
17 |
*Give information separately for cases and controls in case-control studies and, if applicable, for exposed and unexposed groups in cohort and cross-sectional studies.
Note: An Explanation and Elaboration article discusses each checklist item and gives methodological background and published examples of transparent reporting. The STROBE checklist is best used in conjunction with this article (freely available on the Web sites of PLoS Medicine at http://www.plosmedicine.org/, Annals of Internal Medicine at http://www.annals.org/, and Epidemiology at http://www.epidem.com/). Information on the STROBE Initiative is available at www.strobe-statement.org.
- Results: Please Authors consider to shorten “unstandardized coefficient”.
Ans: Thank you for your suggestion. We have shortened to “coefficient β”.
Round 2
Reviewer 3 Report
Authors improved the manuscript as suggested